# Fecal microbiota transplant rescues mice from human pathogen mediated sepsis by restoring systemic immunity

Sangman M. Kim [1,2,3,12], Jennifer R. DeFazio[4,5,12], Sanjiv K. Hyoju[4], Kishan Sangani[1,2], Robert Keskey[1,4], Monika A. Krezalek[4], Nikolai N. Khodarev[6], Naseer Sangwan[4,7], Scott Christley[4], Katharine G. Harris[2], Ankit Malik[1,2], Alexander Zaborin[4], Romain Bouziat[1,2], Diana R. Ranoa[6], Mara Wiegerinck [4], Jordan D. Ernest[2], Baddr A. Shakhsheer[4], Irma D. Fleming [4], Ralph R. Weichselbaum[6], Dionysios A. Antonopoulos[2,7], Jack A. Gilbert [4,7], Luis B. Barreiro[2,8,9], Olga Zaborina[4,13], Bana Jabri[1,2,10,11,13✉] & John C. Alverdy[4,13✉]

Death due to sepsis remains a persistent threat to critically ill patients confined to the intensive care unit and is characterized by colonization with multi-drug-resistant healthcare-associated pathogens. Here we report that sepsis in mice caused by a defined four-member pathogen community isolated from a patient with lethal sepsis is associated with the systemic suppression of key elements of the host transcriptome required for pathogen clearance and decreased butyrate expression. More specifically, these pathogens directly suppress interferon regulatory factor 3. Fecal microbiota transplant (FMT) reverses the course of otherwise lethal sepsis by enhancing pathogen clearance via the restoration of host immunity in an interferon regulatory factor 3-dependent manner. This protective effect is linked to the expansion of butyrate-producing Bacteroidetes. Taken together these results suggest that fecal microbiota transplantation may be a treatment option in sepsis associated with immunosuppression.

[1] Committee on Immunology, University of Chicago, Chicago, IL, USA. [2] Department of Medicine, University of Chicago, Chicago, IL, USA. [3] Department of Biology, University of San Francisco, San Francisco, CA, USA. [4] Department of Surgery, University of Chicago, Chicago, IL, USA. [5] Department of Surgery, Columbia University, New York, NY, USA. [6] Department of Radiation and Cellular Oncology and The Ludwig Center for Metastasis Research, University of Chicago, Chicago, IL, USA. [7] Argonne National Laboratory, Argonne, IL, USA. [8] Department of Genetics, Sainte-Justine Hospital Research Centre, University of Montreal, Montreal, QC, Canada. [9] Department of Pediatrics, Faculty of Medicine, University of Montreal, Montreal, QC, Canada. [10] Department of Pathology, University of Chicago, Chicago, IL, USA. [11] Department of Pediatrics, Section of Gastroenterology, Hepatology and Nutrition, University of Chicago, Chicago, IL, USA. [12] These authors contributed equally: Sangman M. Kim, Jennifer R. DeFazio. [13] These authors jointly supervised this work: Olga Zaborina, Bana Jabri, John C. Alverdy. ✉email: bjabri@bsd.uchicago.edu; jalverdy@surgery.bsd.uchicago.edu

Since the discovery of penicillin in 1928, antibiotics have saved millions of lives, however, their promiscuous use in medicine and agriculture has borne witness to the emergence of multidrug resistant (MDR) pathogens which has become one of the most pressing issues in public health today[1,2]. The most common cause of death in modern intensive care units is "late-onset sepsis," a disorder characterized by colonization with highly virulent MDR pathogens[3]. Colonization of critically ill patients with MDR pathogens is likely a result of the unusual selective pressures to which these patients and their microbiomes are exposed, including life support measures, prolonged antibiotic use, polypharmacy, and artificial feeding. These well-intentioned interventions can result in collapse of the microbiome, the dysregulation of which is increasingly being demonstrated to have a major adverse effect on the immune system. Here we demonstrate that a fecal microbiota transplant (FMT) can rescue mice from lethal sepsis due to a defined four-member pathogen community (PC) isolated from a critically ill patient by reversing the immunosuppressive effect of this PC.

## Results

**FMT rescues mice from gut-derived sepsis**. We infected mice with a PC isolated from the stool of a surgical patient who died of late-onset sepsis, consisting of three species of bacteria (tetracycline resistant *Enterococcus faecalis*, MDR *Klebsiella oxytoca*, and MDR *Serratia marcescens*) and one species of yeast (*Candida albicans*)[4] (Supplementary Table 1). With advances in sequencing, it is now known that the gut is the primary site of colonization and emergence of the "antibiotic resistome" consisting of MDR pathogens and is a major source of lethal systemic infections[5–7]. Therefore, we first inoculated this four-member PC into the gut using a model that exposes the PC to the selective pressures of surgical injury and its treatment. In this model, mice were subjected to 30% hepatectomy, with preoperative food-deprivation and prophylactic antibiotic administration[4,8,9] (Fig. 1a). In accordance with previous studies[9], by postoperative day 1 (POD1) the majority of mice had positive blood, liver, lung, and spleen cultures (Fig. 1b) with clinical signs of sepsis (lethargy, ruffled fur, and chromodacryorrhea) and died by POD2 (Surgery + PC) (Fig. 1c and Supplementary Table 2).

Given that FMT has been shown to drive the competitive exclusion of pathobiota among the indigenous intestinal microbiota[10–12], we applied FMT to our model to determine its ability to rescue mice that would be predicted to develop lethal sepsis. FMT prepared from cecal contents of healthy littermates was administered via enema on 2 consecutive days (Fig. 1a). Remarkably, administration of FMT led to a greater than 70% increase in survival, compared with no improvement in survival when mice received an autoclaved FMT (AC-FMT) (Fig. 1c). The finding that AC-FMT-treated mice had the same clinical phenotype and survival outcome as untreated septic mice indicated that live microbiota needs to be transplanted, and therefore for the remainder of this study, we used AC-FMT treatment as our control for live FMT. Strikingly, FMT treatment resulted in an overall decrease in systemic PC burden by POD2, with significantly lower CFUs of *S. marcescens* in the blood and *C. albicans* in the liver and spleen compared with AC-FMT-treated mice (Mann–Whitney test, $P < 0.05$) (Fig. 1d). Taken together, these results demonstrate that administration of a live, but not an autoclaved FMT, reduces systemic pathogen burden and dramatically increases survival in mice with lethal gut-derived sepsis.

In order to gain further insight into the mechanism of FMT-mediated protection and its ability to reduce systemic PC burden in this model, we performed whole-genome microarray analysis of gene expression in the cecum, liver, and spleen on POD2, the time point where we observed FMT-driven reduction of systemic pathogen burden (Fig. 1d). We looked at genes that were coordinately regulated across all three organs and observed significant changes in gene expression in both FMT and AC-FMT-treated groups compared with non-manipulated control mice (referred to as "Untreated" hereafter) (Fig. 1e and Supplementary Data 1). However, the number of differently expressed genes was approximately tenfold less (349 vs. 3242 at an FDR ≤ 0.01) in FMT-treated mice as compared with AC-FMT-treated septic mice (Fig. 1e and Supplementary Data 1). Among differently expressed genes, the magnitude of the gene expression changes was also consistently lower in FMT-treated mice (Mann–Whitney test, $P < 1 \times 10^{-15}$) (Fig. 1f), suggesting that FMT treatment restores a dysregulated transcriptional program associated with PC dissemination. Pathway and database analysis of the PC-induced dysregulated transcriptional program (Supplementary Fig. 1, Supplementary Data 1, and 2) suggested that the systemic dissemination of pathogens was accompanied by a coordinated abrogation of signaling downstream of pattern recognition receptors (PRRs), a pathway that is commonly targeted by microbial pathogens and required for their clearance[13–17]. Both interferon regulatory factor 3 (IRF3) and the nuclear factor-kappa B (NF-κB) signaling pathways were dysregulated in AC-FMT-treated septic mice, while FMT treatment restored these pathways back to homeostatic levels in three different organs, two of which are remote from the gut (Supplementary Fig. 1a, Supplementary Data 1, and 2).

Intriguingly, although the translocation of the PC began before treatment with FMT on POD1 (Fig. 1b), FMT was still able to drive a reduction in PC burden in systemic organs (Fig. 1d) suggesting two possible, but non-mutually exclusive mechanisms. First, FMT could promote colonization resistance and the competitive exclusion of the PC within the gut environment in a manner similar to the mechanism by which FMT clears recurrent *Clostridium difficile* infection[10–12] thereby preventing further translocation and facilitating systemic clearance of the PC. Indeed, temporal sequencing of 16S rRNA of cecal contents demonstrated that FMT, but not AC-FMT treatment, restored intestinal microbial diversity (Supplementary Fig. 2a–d and Supplementary Table 3) in association with a marked reduction of all three inoculated bacterial pathogens over time (Supplementary Fig. 2e–g). By POD7, the inoculated pathogens were completely absent from the ceca of FMT-treated mice (Supplementary Fig. 2e–g). A second mechanism might involve the ability of FMT to drive a recovery-directed immune response at the systemic level, as suggested by our transcriptional analysis (Fig. 1e,f, Supplementary Fig. 1, Supplementary Data 1 and 2), thereby enhancing bacterial clearance in peripheral organs and preventing the progression to lethal infection.

**FMT drives the clearance of systemically disseminated PC**. To test the latter possibility that FMT drives a systemic recovery-directed immune response in the above model, we injected the PC directly into the peritoneal cavity (intraperitoneal-i.p.) which caused immediate systemic dissemination of pathogens; in this series of experiments there was no fasting, surgery, or antibiotics (IP model) (Fig. 2a). Therefore, this model separated the FMT treatment and PC inoculation into physically isolated compartments (i.e., FMT via enema, PC via i.p. injection) thus allowing assessment of the direct role of FMT on systemic PC clearance. In this model, PC inoculation resulted in immediate systemic sepsis and death in more than 60% of mice in less than 72 h (PC) (Fig. 2b). To assess the ability of an FMT enema to reverse the mortality from i.p. administered PC, mice were treated with

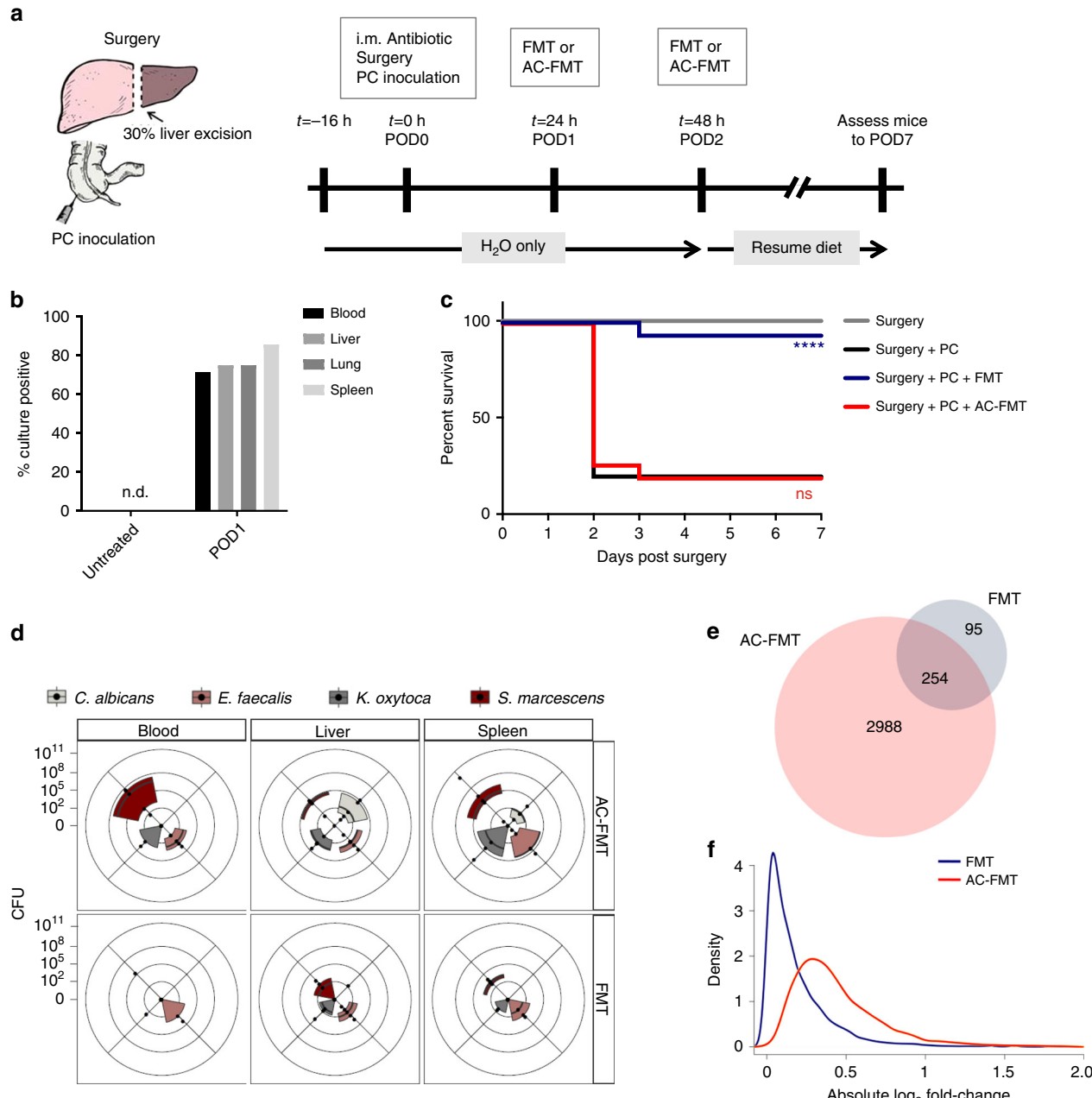

**Fig. 1 FMT rescues mice from lethal infection due to a four-member pathogen community isolated from a patient with lethal sepsis. a–f** To mimic the development of sepsis following major surgery, C57BL/6 mice were starved and received a single-intramuscular injection of cefoxitin, after which they were opened, subjected to 30% hepatectomy, and inoculated with the PC directly into their cecum. To test the protective effects of FMT, 24 h (POD1) and 48 h (POD2) after surgery, FMT or AC-FMT was administered via enema. **a** Sketch and timeline of the gut-derived sepsis model. **b** The percentage of mice with microbes present on POD1 in the indicated sites was measured by homogenizing organs and culturing them on TSB agar plates. The percentage of mice that had positive cultures out of all of the mice assessed are graphed for the groups indicated ($n = 5$, 7, and 8 mice for untreated, POD1 blood and spleen, and POD1 liver and lung, respectively). **c** Kaplan–Meier survival curves for indicated groups. "Surgery" are control mice that had starvation, antibiotics, and surgery without pathogen inoculation ($n = 15$ mice/group; Log-rank (Mantel–Cox) test; $P < 0.0001$ between Surgery + PC/Surgery + PC + FMT, $P = 0.9554$ between Surgery + PC/Surgery + PC + AC-FMT, $P < 0.0001$ between Surgery + PC + FMT/Surgery + PC + AC-FMT). **d** Quantitative culture results of indicated sites comparing the PC burden of Surgery + PC + FMT and Surgery + PC + AC-FMT-treated mice at POD2. Each of the four PC members were assessed separately using selective culture plates. On radar plot, each dot along the axes represents the CFU/mL of blood or CFU/mg of liver and spleen of one mouse, the further the distance from the center of the plot, the higher the pathogen burden. The third and first quartiles and the median are indicated with the edges of the colored boxes and the thick black line respectively ($n = 5$ mice/group; Mann–Whitney test; PC burden significantly different between AC-FMT/FMT in the blood [SM*], liver [CA*], spleen [CA*], *$P \leq 0.05$). **e, f** The cecum, liver, and spleen of Surgery + PC + FMT or Surgery + PC + AC-FMT-treated mice was harvested on POD2, mRNA was extracted from whole tissues, and gene expression was measured using whole-genome microarray. Gene expression was compared against the baseline of untreated control mice (denoted "Untreated"; $n = 3$ mice/group). **e** Venn diagram showing the number of genes differently expressed (FDR $\leq 0.01$) across all organs on POD2 in FMT and AC-FMT-treated mice. **f** Distribution of the absolute log2-fold change in FMT and AC-FMT-treated mice on POD2 among genes differently expressed in either condition (Mann–Whitney test; $P < 1 \times 10^{-15}$).

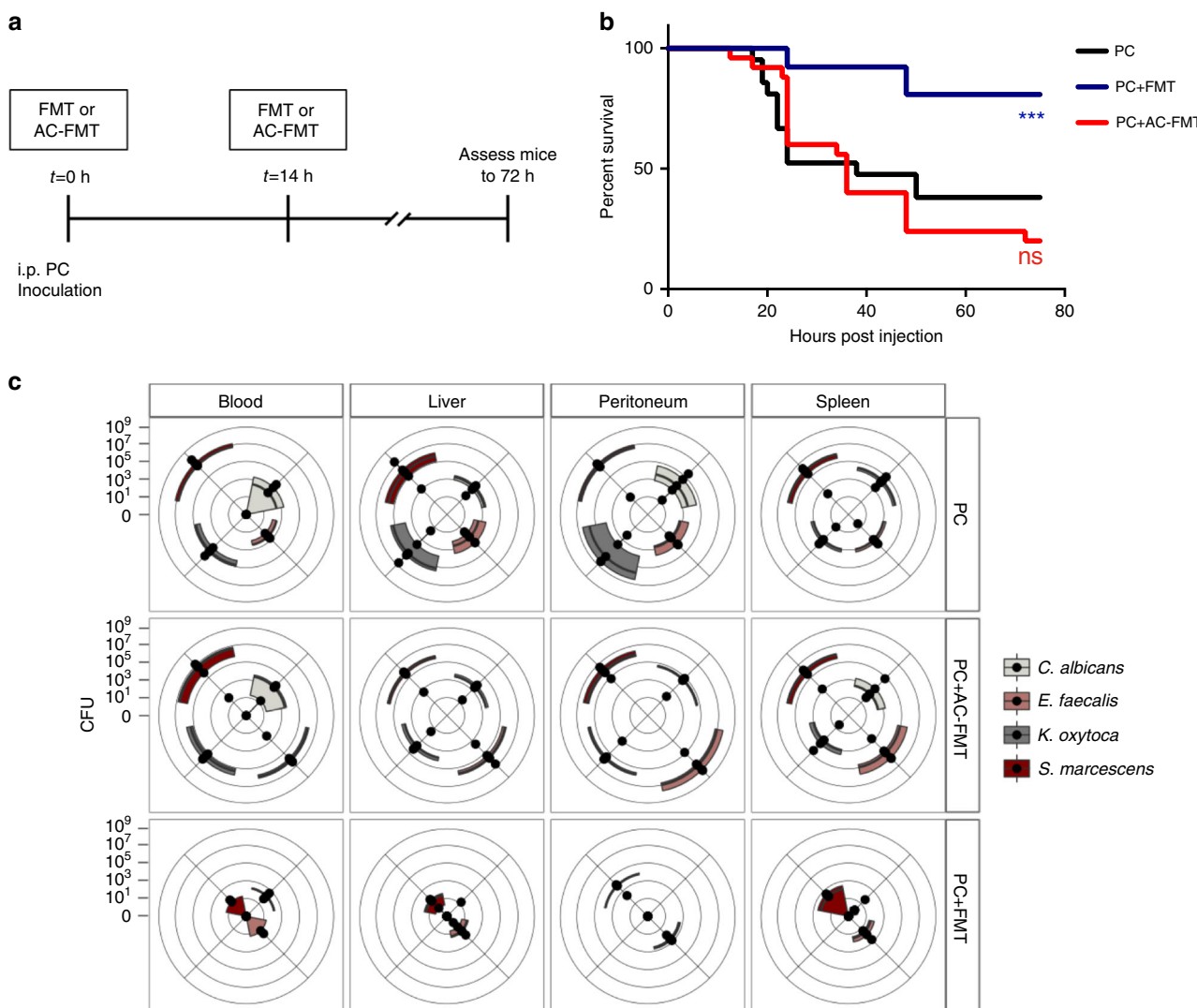

**Fig. 2 FMT drives the clearance of systemically disseminated pathogens. a–c** In order to cause immediate systemic dissemination, the PC was injected directly into the peritoneum (IP) of mice. In the IP model, mice were not subjected to starvation, antibiotic treatment, or hepatectomy. To test the protective effects of FMT in this model, immediately upon injection of PC and then once more 14 h post injection, FMT or an AC-FMT was administered via enema. **a** Timeline of the IP sepsis model. **b** Kaplan–Meier survival curves ($n = 21$, 26, and 25 for PC, PC + FMT, and PC + AC-FMT-treated mice, respectively; Log-rank (Mantel–Cox) test; $P = 0.001$ between PC/PC + FMT, $P = 0.4905$ between PC/PC + AC-FMT, $P < 0.0001$ between PC + FMT/PC + AC-FMT). **c** Quantitative culture results of indicated sites comparing the PC burden of PC, PC + FMT, and PC + AC-FMT mice ~20 h post injection of PC. On radar plot, each dot along the axes represents the CFU/mL of blood and peritoneal fluid or CFU/mg of liver and spleen of one mouse, the further the distance from the center of the plot, the higher the pathogen burden. The third and first quartiles and the median are indicated with the edges of the colored boxes and the thick black line respectively ($n = 5$ mice/group); Kruskal–Wallis test/Dunn's multiple comparisons test; PC burden significantly different between PC/PC + FMT in the blood [SM*], liver [KO**, SM*], peritoneum [CA*, KO*, SM*], and spleen [CA**, KO*]. PC burden significantly different between PC + AC-FMT/PC + FMT in the blood [EF**, KO*], liver [CA**, EF**, SM*], peritoneum [CA**, EF**, KO*], and spleen [EF*, KO*, SM*]. No significant differences between PC/PC + AC-FMT. *$P \leq 0.05$, **$P \leq 0.01$.

either FMT or AC-FMT via enema two times; at the time of PC inoculation and 14 h post inoculation (Fig. 2a). Remarkably, administration of FMT resulted in a greater than 40% increase in survival in the i.p. injected mice compared with no improvement in survival in AC-FMT-treated mice (Fig. 2b). Treatment with FMT resulted in significant clearance of all the PC members from peripheral organs compared with AC-FMT-treated mice (Fig. 2c). Taken together these data suggest that FMT can modulate systemic host physiology in a way that drives pathogen clearance at remote sites.

**The rescue effect of FMT occurs in an IRF3-dependent manner.** Microbial pathogens are sensed by the wide array of PRRs

that are found on the surface and inside of cells that lead to the activation of the NF-κB and/or IRF3 signaling pathways. In the case of Gram-negative bacteria, recognition by TLR4 causes the concomitant activation of both pathways[18,19]. Many successful pathogens have evolved methods to avoid detection by these PRRs through a variety of molecular mechanisms[13–17], including the transcriptional modulation of host signaling pathways[17,20,21]. Consistent with our findings in the gut-derived lethal infection model (Fig. 3a, Supplementary Fig. 1, Supplementary Data 1 and 2), i.p. injection of the PC resulted in downregulation of *IRF3* and the upregulation of *NF-kB Inhibitor Alpha (NFKBIA), NF-kB Inhibitor Beta (NFKBIB),* and *TNF Alpha Induced Protein 3 (TNFAIP3),* known NF-κB pathway inhibitors[22,23], in the cecum,

liver, and spleen, 20 h post PC injection in AC-FMT mice (Fig. 3b, Supplementary Fig. 3a, and Supplementary Table 4). Intriguingly, we observed the same gene expression changes in vitro when we cultured mouse embryonic fibroblasts (MEFs) with individual members of the PC that were either live (Supplementary Fig. 3b) or lysed and filtered (Fig. 3c and Supplementary Fig. 3c), with specific members driving more dramatic levels of transcriptional modulation. In particular, both live *S. marcescens* and its lysates drove the most dramatic downregulation of IRF3, while live *K. oxytoca* drove the most dramatic upregulation of NF-κB inhibitors (Fig. 3c and Supplementary Fig. 3b, c). Bacterial lysates prepared from cecal contents of untreated mice (Microbiota) did not lead to a reduction in *IRF3*, although it did upregulate *NFKBIA* and *TNFAIP3* at the highest dose tested (Fig. 3c and Supplementary Fig. 3c). Taken together, these results suggest that individual members of the PC have both distinct and complementary abilities to directly modulate NF-κB and IRF3 signaling with the net result being blockade of NF-κB and IRF3 signaling in both models of infection. Importantly, in both models, treatment with FMT restored these signaling pathways back to homeostatic levels (Supplementary Fig. 1a, Fig. 3b, and Supplementary Fig. 3a) suggesting that FMT may protect these mice through restoration of immune homeostasis in the face of direct immune subversion by the PC.

IRF3 is a transcription factor that drives the production of type I interferons and inflammatory cytokines[18,19], and intact IRF3 signaling is required for immunity against certain Gram-positive and Gram-negative bacteria[24–26]. The observation that MyD88-deficient patients are resistant to most common bacterial infections[27] further suggests that the IRF3 pathway, which is not dependent on MyD88, may be involved in protective immunity. Importantly, unlike mice that lack NF-κB signaling components[28], IRF3-deficient mice do not display overt developmental or immunological defects at steady-state conditions[29]. Finally, in both the gut-derived and IP models of sepsis herein described, IRF3 was observed to be downregulated with the progression of sepsis across organs, and treatment with FMT restored IRF3 to normal levels (Supplementary Fig. 1a and Fig. 3b). Therefore, we hypothesized that IRF3 may be required for FMT-mediated protection. To test this, we performed reiterative experiments in the IP model with IRF3-sufficient and IRF3-deficient littermates. In accordance with IRF3 playing a role in protective immunity against the PC, IRF3-deficient mice displayed increased mortality upon PC infection as compared with IRF3-sufficient littermates, with mice heterozygous for IRF3 having intermediate survival (Supplementary Fig. 4). Importantly, IRF3-deficient mice did not demonstrate an increase in survival upon treatment with FMT (Fig. 3d) and FMT-treated IRF3-deficient mice had significantly higher PC burden in peripheral sites when compared with FMT-treated IRF3-sufficient littermates (Fig. 3e) indicating that FMT required intact IRF3 to clear the PC and rescue mice from pathogen-induced lethal sepsis. These results are in contrast to findings reported with the cecal puncture and ligation model of sepsis in which commensal microbiota and a grossly ischemic cecum drive the process of lethal sepsis. In the cecal ligation model, IRF3-deficiency has been shown to be protective[30], potentially indicating the difference between sepsis driven by mouse commensal microbiota versus sepsis caused by a defined community of sepsis-associated human pathogens.

**PC infection drives a reduction in fecal butyrate levels**. We next sought to clarify how FMT might protect mice in the IP model, given that, unlike in the gut-derived sepsis model, it is presumed that IP injected mice will have maintained an intact microbiota at the time of FMT treatment. However, others have shown that any

sudden physiologic insult can rapidly (within 6 h) deplete microbiome composition and function[31]. I.p. injected mice demonstrated no significant change in either total bacterial load (Supplementary Fig. 5a) or microbiota composition (Supplementary Fig. 5b, c) at 14 h post PC injection. This lack of compositional change by 16S rRNA analysis was also observed in mice that received AC-FMT or FMT treatment (Supplementary Fig. 5a–c). This finding further indicates that FMT in this model does not mediate survival by promoting colonization resistance and the competitive exclusion of the PC.

Healthy microbiota can affect systemic immune homeostasis and enhance pathogen clearance via the production of metabolites such as short chain fatty acids (SCFA)[32–34]. To further define through which mechanisms FMT may mediate its effect, we investigated whether there were any alterations in SCFA expression in PC injected septic mice. Strikingly, cecal butyrate levels, but not acetate and propionate levels (Supplementary Fig. 6a, b), were significantly diminished in mice following i.p. PC injection (Fig. 4a). Consistent with this observation, selective downmodulation of butyrate production has been reported in other models of inflammation[35]. These findings led us to investigate whether reduction and restoration in butyrate levels could be linked to changes in specific bacterial populations (OTUs).

Interestingly, in PC infected mice, a significant reduction in the abundance of five OTUs from the phylum Firmicutes are members of the *Clostridium* cluster XIVa (based on RDP SeqMatch searches[36]; Supplementary Fig. 6c), which is a cluster that contains many taxa that are known butyrate producers[37]. Butyrate kinase (*buk*) and butyrate transferase (*but*) are the enzymes catalyzing the last step in butyrate production by bacteria[37]. More specific analysis, however, found only one of the OTUs proportionally decreased in PC infected mice to match organisms containing the *buk* or *but* genes in the RDP FunGene database[36]. Hence, it is possible that in addition to changes in OTUs, there is a defect in the transcription of butyrate processing enzymes, and/or an increased uptake of butyrate that contribute to the observed decreased expression in butyrate. Whatever caused the decrease in butyrate, FMT treatment of septic mice led to a restoration of butyrate levels (Fig. 4a) that was associated with a significant increase in OTUs that belonged to the phylum Bacteroidetes, including a large number of S24-7 family members. Based on RDP FunGene searches most of these OTUs have *buk* and/or *but* (Fig. 4b). Although there was some overlap between untreated mice and septic mice receiving FMT (Supplementary Fig. 6d and Supplementary Data 3), the OTUs from Bacteroidetes are specifically enriched in PC + FMT mice (Supplementary Fig. 6d). There are indications that members of Bacteroidetes actively alter the gut environment to their benefit[38] and are more resilient colonizers than other phyla in the presence of acute inflammation[39]. The rapid colonization of septic mice by "non-traditional" butyrate-producing members of Bacteroidetes may hence account for the restoration of butyrate levels in FMT-treated mice.

**Butyrate and HDACi normalize IRF3 levels in vitro**. Given that butyrate has histone deacetylase inhibitory (HDACi) activity that can positively or negatively modulate transcription, we hypothesized that FMT may mediate its beneficial effects via the HDACi properties of butyrate. To test this hypothesis, we determined whether butyrate could restore normal levels of IRF3 in vitro. When MEFs were treated with an equal concentration of lysates from all four members of the PC, IRF3 transcript levels were reduced to ~50%; however, culturing these cells in the presence of butyrate led to a restoration of IRF3, in a

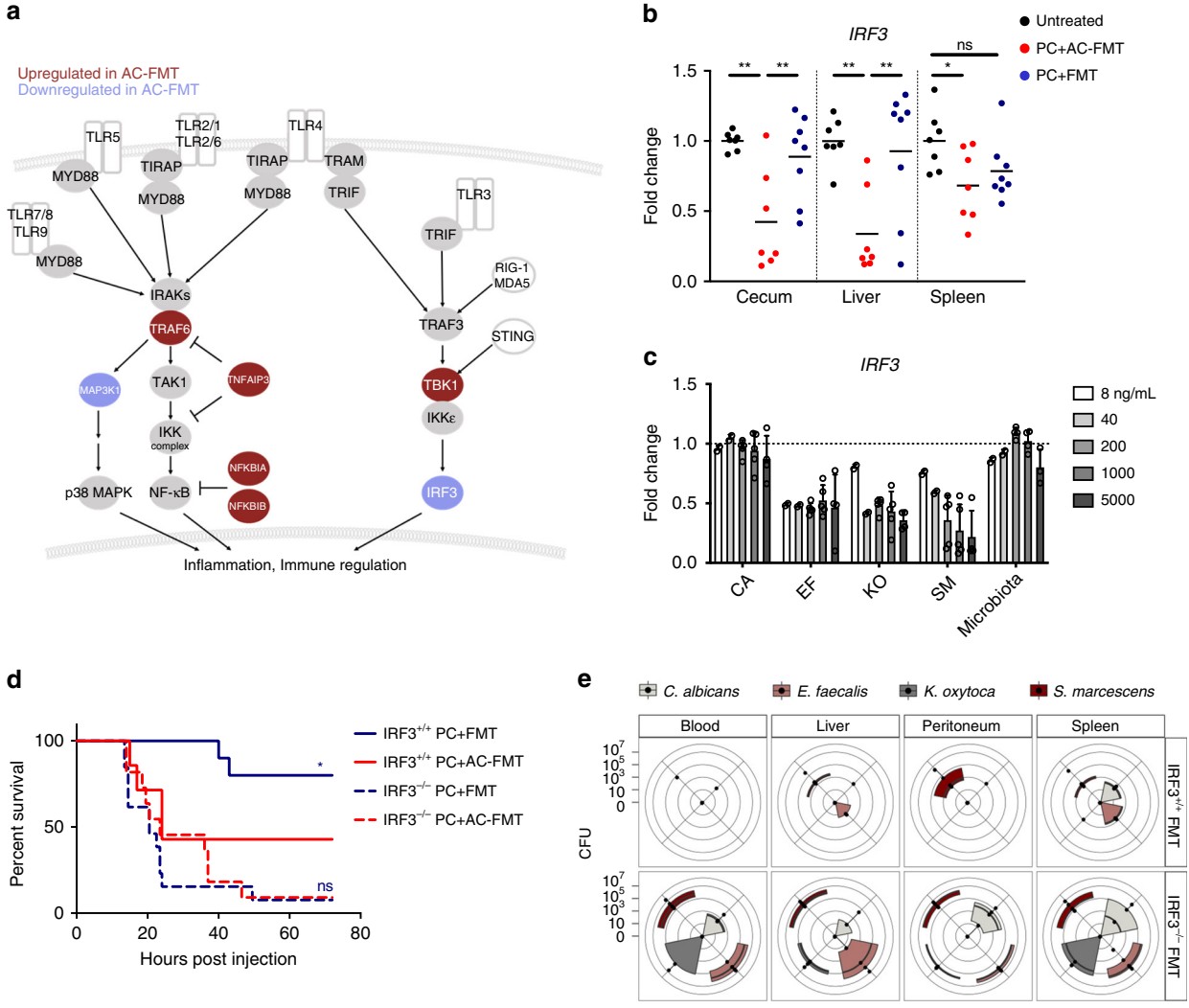

**Fig. 3 Clearance of systemically disseminated pathogens by FMT requires intact IRF3. a** Summary of transcriptional changes in the NF-κB/IRF3 signaling pathway downstream of various pattern recognition receptors in the cecum, liver, and spleen of PC + Surgery + AC-FMT mice on POD2. Red shaded circles are genes that are significantly upregulated, blue shaded circles are genes that are significantly downregulated. **b** Mice were injected i.p. with PC and treated with FMT or AC-FMT as before. RNA was isolated from indicated organs ~20 h post injection of PC and qPCR for *IRF3* was performed. (Fold change compared with the mean of untreated group *IRF3* relative expression for each organ; $n$ = 7, 7, and 8 for untreated, PC + AC-FMT, and PC + FMT-treated mice, respectively; center is mean; one-way ANOVA/Tukey's multiple comparison; *$P \leq 0.05$, **$P \leq 0.01$). **c** MEFs were cultured for 12 h in culture media containing filtered lysates made from cultured individual PC members, or from cecal contents of untreated mice at the indicated concentrations. MEF RNA was isolated and *IRF3* expression was measured by qPCR. (Fold change compared with the baseline of untreated MEF *IRF3* relative expression [highlighted with the dotted line at $y$ = 1] shown; 3 independent experiments; $n \geq 3$ for all conditions; mean + SD). **d, e** To assess the requirement for IRF3 in FMT-mediated protection, IRF3$^{+/+}$ and IRF3$^{-/-}$ littermates were injected i.p. with PC and treated with FMT or AC-FMT as before. **d** Kaplan–Meier survival curves for genotypes and treatments indicated ($n$ = 10, 7, 13, and 11 for IRF3$^{+/+}$ mice with PC + FMT, IRF3$^{+/+}$ mice with PC + AC-FMT, IRF3$^{-/-}$ mice with PC + FMT, and IRF3$^{-/-}$ mice with PC + AC-FMT, respectively; Log-rank (Mantel–Cox) test; $P$ = 0.0469 between IRF3$^{+/+}$ PC + FMT/IRF3$^{+/+}$ PC + AC-FMT, $P$ = 0.5440 between IRF3$^{-/-}$ PC + FMT/IRF3$^{-/-}$ PC + AC-FMT). **e** Quantitative culture results of indicated sites comparing the PC burden of PC + FMT-treated IRF3$^{+/+}$ and IRF3$^{-/-}$ mice ~20 h post injection of PC. On radar plot, each dot along the axes represents the CFU/mL of blood and peritoneal fluid or CFU/mg of liver and spleen of one mouse, the further the distance from the center of the plot, the higher the pathogen burden. The third and first quartiles and the median are indicated with the edges of the colored boxes and the thick black line respectively ($n$ = 5 mice/group; Mann–Whitney test; PC burden significantly different between IRF3$^{+/+}$ PC + FMT/IRF3$^{-/-}$ PC + FMT in the blood [EF*, SM*], liver [EF*, KO*, SM**], peritoneum [CA*, KO*, SM*], and spleen [EF*, KO*]. *$P \leq 0.05$, **$P \leq 0.01$).

dose-dependent manner (Fig. 4c). The restoration of IRF3 transcript levels by butyrate was dependent on the HDACi activity of butyrate, and not on the effect of butyrate to signal through G-protein coupled receptor 109A (GPR109a), as valproate (HDAC Class I and IIa inhibitor) but not niacin (GPR109a agonist)[32] was able to restore IRF3 transcript levels in vitro (Fig. 4c). The SCFA propionate is also known to have HDACi activity, while acetate does not[32], and thus propionate, but not acetate, was also able to

restore IRF3 (Fig. 4c). This is in line with previous studies that have demonstrated a protective effect of both butyrate and HDACi in animal models of sepsis[40–44]. We cannot eliminate that in addition to restoring IRF3 levels, the presence of higher levels of butyrate may also have a direct cytotoxic effect on PC[45,46]. Furthermore, metabolites other than butyrate may contribute to the clearance of PC in vivo[47]. Taken together, these data suggest that critical illness itself may result in the depletion of

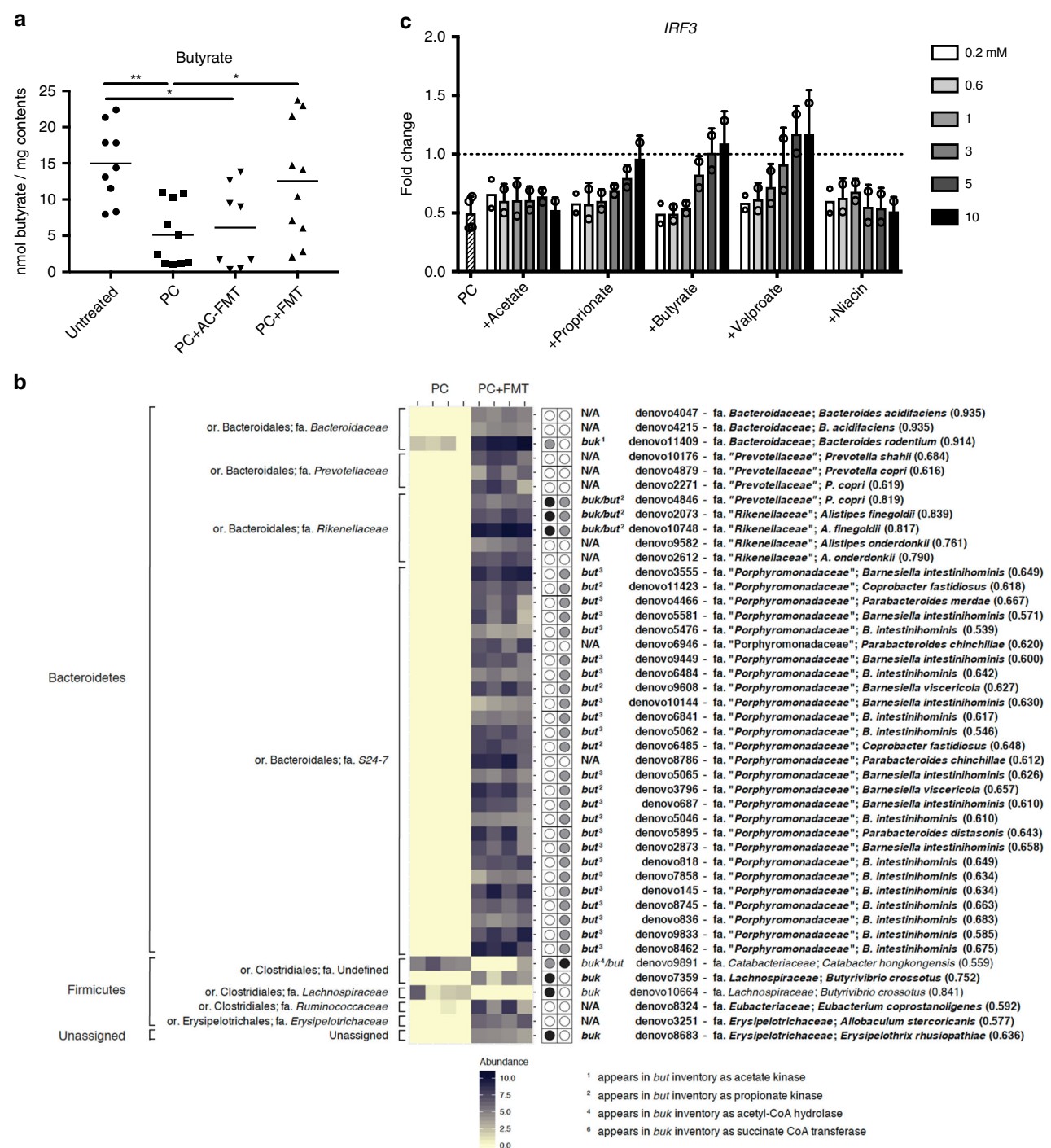

intestinal butyrate. FMT, by providing OTUs such as Bacteroidetes, that can expand under inflammatory conditions and produce butyrate, is able to protect against systemic PC infection.

## Discussion

Here we present evidence that a PC expressing varying degrees of antibiotic resistance isolated from the stool of a terminally septic patient dysregulates, in a concerted manner, the transcriptional program associated with host defenses against infections. By blocking key parts of both the NF-κB and IRF3 signaling pathways, these pathogens prevent the host from mounting a protective immune response, which likely accounts for the high systemic pathogen burden, host deterioration, and death seen in

this model. Unexpectedly, we found that FMT was able to counter the immunosuppressive effects of these pathogens in part through its ability to restore normal levels of butyrate. The ability of FMT to provide protection in the IP model of lethal infection indicates that FMT can act systemically, beyond its local effect on the gastrointestinal microbiota or pathobiota, and restore protective immunity against highly virulent and MDR pathogens. Understanding how FMT drives a recovery-directed immune response at the systemic level may have important implications for the management of critically ill patients who are regularly treated with antibiotics and colonized by healthcare-associated pathogens. Given that the pathogen communities that colonize critically ill patients each carry their own unique life histories, results from this study underscore the importance of humanizing the

**Fig. 4 Pathogen infection drives butyrate deficiency and butyrate can restore IRF3 in vitro. a** Mice were injected IP with PC and treated with FMT or AC-FMT as before. Cecal contents were collected ~20 h post injection of PC and butyrate levels were measured by GC–MS. (One-way ANOVA/Tukey's multiple comparison, *$P \leq 0.05$, **$P \leq 0.01$). **b** Analysis of the OTUs found in the datasets reveals that FMT increases the presence and abundance of butyrate-producing bacteria. Changes in specific OTUs between mice treated with or without FMT were detected using the R packages phyloseq and DESeq2[62,63]. The figure depicts significant log2-fold changes based on an FDR (Benjamini–Hochberg) significance threshold of 0.01 in the relative abundance of specific OTUs between PC and PC + FMT mice as heatmaps. Representative sequences for each of these OTUs were then used to search the Ribosomal Database Project's set of quality-controlled, aligned, and annotated bacterial 16S rRNA sequences using SeqMatch[36] in order to obtain nearest neighbor matches with cultured representatives at the genus-species level. On the left are the phylum and family classifications of the OTUs and on the right are the putative genus/species designation (numbers in parentheses indicate SeqMatch score [Sab]). The panel to the right of the heatmaps indicate whether butyrate kinase (*buk*) or butyryl-CoA:acetate CoA-transferase (*but*) encoding genes have been detected in the genomes of the SeqMatch genus/species assignments based on the *buk* and *but* entries in the RDP FunGene functional gene pipeline and repository[64]. A closed black circle indicates the presence of a *buk* or *but* gene in that genome, whereas a closed gray circle indicates an alternate annotation (defined at the bottom of the figure). An open circle indicates that no match was found in the RDP FunGene. Those listed in bold on the right are enriched in the PC + FMT mice. **c** MEFs were treated with of equal concentrations of filtered lysates from all four PC members (250 ng/mL) in the absence (striped) or presence of indicated amounts of compounds for 12 h. MEF RNA was isolated and qPCR was performed to measure the effect of PC members on *IRF3* expression in vitro (fold change compared with the baseline of untreated MEF *IRF3* relative expression [highlighted with the dotted line at $y = 1$] shown; three independent experiments; $n \geq 3$ for all conditions; mean + SD).

mouse gut with clinically relevant human pathogens when assessing the immune response. The immunosuppressive nature of the human PC herein described may explain, in part, why mouse models of endotoxin administration or sepsis due to rodent flora have failed to translate into effective therapies in clinical trials[48,49]. Whether the current approach will be useful in the design of future human trials will require further study.

## Methods

**Mice**. Seven-to-nine-week-old-male C57BL/6 mice weighing 18–22 g were used for experiments. WT C57BL/6 were purchased from Charles River Laboratories. IRF3$^{-/-}$ mice on a C57BL/6 background were kindly provided by Dr Tatyana V. Golovkina (University of Chicago) and mice used in comparative studies were the progeny of IRF3$^{+/-}$ mice resulting in IRF3$^{+/+}$ and IRF3$^{-/-}$ littermates. IRF3$^{+/+}$ and IRF3$^{-/-}$ littermates were cohoused until the day of the experiment to prevent microbiome related differences between genotypes from affecting results.

**PC preparation**. The PC consisted of four members including *C. albicans*, *E. faecalis*, multidrug-resistant *K. oxytoca*, and multidrug-resistant *S. marcescens* and was isolated from the stool of a critically ill patient (ICU1-2) who was exposed to numerous antibiotics during the course of their sepsis[4]. The individual pathogens were plated on TSB agar from individual frozen stocks and grown overnight at 37 °C. Colonies were suspended in liquid TSB medium, subgrown for 1 h and then adjusted to an optical density measured at wavelength of 600 nm (OD600) of 0.2. All four microbe species were then combined together in equal volumes. The microbial suspension was centrifuged at 6000 rpm for 10 min, the excess TSB removed and the remaining pellet was resuspended in the same volume of 10% glycerol. The resulting microbial suspension was administered in the two sepsis mouse models as described below.

**Gut-derived sepsis mouse model**. All experiments were approved by the Institutional Animal Care and Use Committee at the University of Chicago (IACUC protocol 71744). Mice were routinely fed tap water and Harland Teklad feed (Madison, WI) under 12-h light/dark cycles and were allowed to acclimate for at least 48 h before surgery. In order to mimic preoperative surgical conditions, mice were exposed to food restriction and prophylactic antibiotics. Approximately 16 h prior to surgery, mice were moved to wire floor cages to prevent coprophagy. Food was removed, and mice were allowed only tap water ad libitum. Each mouse received an intramuscular cefoxitin injection 30 min prior to the incision at a concentration of 25 mg/kg into the left thigh. Mice were then subjected to a midline laparotomy, 30% hepatectomy of the left lateral lobe of the liver and a direct cecal inoculation of PC (200 μL of PC suspension prepared as described above). On the day of surgery, a fresh FMT was prepared by harvesting the cecal contents of normal healthy mice. Cecal contents were then suspended in sterile 10% glycerol at a concentration of 50 mg/mL. An aliquot of the cecal contents was autoclaved and resuspended in sterile 10% glycerol to a final concentration of 50 mg/ml to create the autoclaved control (AC-FMT). On postoperative days one and two (POD1 and POD2), randomly selected group of mice were given either 1 ml of FMT or AC-FMT via rectal enema using a blunt vein irrigation cannula (Shiley/Pfizer). Chow was restored to the cages on POD2, ~45 h after the operation. Survival and health status were monitored and documented for 7 days using the clinical scoring system shown in Supplementary Table 2 (after 7 days, mice that survive until this point,

regardless of treatment group return to full health and survive indefinitely). All mice that were moribund were sacrificed and considered to be a mortality.

**Nonquantitative culture of PC**. The blood, liver, lung, and spleen of mice were harvested on POD1 to determine the percentage of mice that had microbes present in these compartments. Blood was collected via direct cardiac puncture, mixed with glycerol to a final concentration of 10%, and stored at −80 °C. Ten milligrams of liver, lung, and spleen were homogenized in 500 μl of 10% glycerol and stored at −80 °C. Each sample was serially diluted and plated on TSB agar plates. All the plates were incubated in 37 °C incubator for 24 h and then assessed for the presence of colonies.

**Quantitative culture of PC**. The blood, spleen and liver of mice were assessed for colonization with PC species using selective plates: for *C. albicans*, CHROMagar Candida plates (BBL); for *K. oxytoca*, MacConkey supplemented with ciprofloxacin, 10 μg/ml; for *S. marcescens*, MacConkey supplemented with imipenem, 1 μg/ml; and for *E. faecalis*, Enterococcal agar plates (BBL) supplemented with tetracycline, 100 μg/ml. The bacterial plates supplemented with antibiotics were chosen based on their antibiotic resistance[4] (Supplementary Table 1), and the selectivity was verified by plating intestinal content of untreated mice with no growth on the plates. CHROMagar Candida plates select for *C. albicans* seen as blue colonies. Enterococcal agar plates select for *Enterococcus* seen as black colonies. Blood, liver, and spleen were collected as before and stored at −80 °C. Peritoneal fluid was collected by 1 mL syringe and stored at −80 °C. Each sample was serially diluted, and 50 μl of each dilution was plated on above mentioned four selective plates. All the plates were incubated in 37 °C incubator for 24 h. CFU counts were normalized to tissue weight for spleen and liver and volume for blood and peritoneal fluid.

**Mouse genome transcription analysis**. As indicated in the original model description, mice underwent the full protocol to allow for the development of gut-derived sepsis (i.e., intestinal inoculation of the PC and surgical hepatectomy) and were then treated with either a live FMT or AC-FMT ($n = 3$/group).

We performed gene expression analysis on POD2, because this was the time point where we observed FMT-driven reduction of systemic pathogen burden. In accordance with this reduction in systemic pathogen burden, the FMT-treated mice had a clinical score of 1, while the AC-FMT-treated mice had a clinical score of 4 at the time of sampling (scoring system described in Supplementary Table 2). The cecum, left lobe of the liver, and spleen were resected and stored in RNA later (Qiagen). Tissue was homogenized using Tissue-Tearor (Biospec Products) and RNA extraction was performed using the RNeasy Plus Mini Kit (Qiagen) according to the manufacturer's instructions. Whole-genome transcriptional profiling was performed using Illumina MouseRef-8 v2 arrays (Illumina) by the Functional Genomics Facility of the University of Chicago.

Low-level microarray analyses were performed in R, using the Bioconductor software package lumi[50]. We first applied a variance stabilizing transformation to all arrays[51] and then quantile normalized the data. After normalization, we removed probes with intensities indistinguishable from background noise (as measured by the negative controls present on each array). After these preprocessing steps, data from 13,312 genes were available for differential expression analysis.

To identify genes whose expression levels were altered at POD2 after AC-FMT or FMT treatment (as compared with untreated control mice), we used a linear modeling-based approach. Specifically, we used the Bioconductor limma package[52] to fit, for each gene, a linear model with individual treatment (i.e., AC-FMT or FMT) and tissue as fixed effects. To identify changes in gene expression within each tissue we ran a nested linear model where treatment was nested within tissue type.

For each gene, we subsequently used the empirical Bayes approach of Smyth[40] to calculate a moderated t statistic and P value. We corrected for multiple testing using the false discovery rate (FDR) approach of Benjamini and Hochberg[53]. Gene Ontology enrichment analysis were done using Gorilla[54] using all expressed genes (i.e., 13,312 genes) as a background set.

**16S rRNA analysis.** Microbial DNA was extracted from cecal luminal contents and cecal tissues using Bacteremia DNA Isolation Kit (BiOstic, 12240-50), and the 16S rRNA analysis was performed at Argonne National Laboratories (Lemont, IL). Samples were analyzed by 16S rRNA V4 iTAG amplicon sequencing analysis. Paired end reads were quality trimmed and processed for OTU (operational taxonomic unit) clustering using UPARSE pipeline[55], set at 0.97% identity cutoff. Taxonomic status was assigned to the high quality (<1% incorrect bases) candidate OTUs using the "parallel_assign_taxonomy_rdp.py" script of QIIME software[56]. Multiple sequence alignment and phylogenetic reconstruction was performed using PyNast and FastTree[56]. OTU matrix was processed to remove OTUs containing less than five reads in order to reduce the PCR and sequencing based bias; then the OTU table was rarified to the minimum numbers of reads present in the smallest library. We used the oligotyping pipeline[57] to identify the sub-OTU level differences in the introduced pathogen strains of K. oxytoca ICU1-2, S. marcescens ICU1-2, and E. faecalis ICU1-2. Full length 16S rRNA gene sequence was extracted from genome sequences of K. oxytoca, S. marcescens, and E. faecalis genomes using Blastn. Megablast (minimum identity cutoff = 100%) was used to confirm the strain identification between full length 16S rRNA gene sequences and oligotype representative sequences.

**Sequencing of bacterial pathogen genomes.** For isolation of DNA, strains of the original stock consisting of K. oxytoca, S. marcescens, and E. faecalis were collected from exponential phase during growth in liquid TSB. Sequence libraries were generated using Nextera XT protocol according to manufacturer's instructions (Illumina). Libraries were sequenced by whole-genome shotgun sequencing using the Illumina HiSeq system at Argonne National Laboratory (Lemont, IL). Approximately 100-fold coverage of each genome was generated. Reference-guided genome assembly was conducted using a combination of de novo assembly and read alignment to the K. oxytoca CAV1374 (NZ_CP011636.1), S. marcescens CAV1492 (NZ_CP011642.1), or E. faecalis V583 (NC_004668.1) genome. De novo assembly was performed using the Spades version 3.5.0[58] for each species. Reads were aligned to the appropriate species-specific reference genome using Bowtie2 version 2.2.5[59]. Contigs from de novo assembly, greater than 1000 bp in length, were placed onto the reference genome using BLAST version 2.2.30[60], and merged into longer scaffolds with reference-aligned sequence using custom Perl scripts. The draft genome sequence of K. oxytoca ICU1-2b is 5,852,736 bp in length across 12 scaffolds, S. marcescens ICU1-2a is 5,115,171 bp in length across 20 scaffolds and E. faecalis ICU1-2c is 2,929,980 bp in length across 14 scaffolds. Accession numbers in NCBI are LQAM00000000 Enterococcus, LQAL00000000 Klebsiella, and LQAK00000000 Serratia in the BioProject PRJNA307050.

**Intraperitoneal (IP) sepsis mouse model.** In order to determine if the rescue effect of FMT functions at the level systemic infection, we directly injected mice with the PC into the peritoneum (i.p.). In this IP model, mice were not subjected to starvation, antibiotic treatment, or hepatectomy. The suspension was prepared using the identical strains (S. marcescens, C. albicans, K. oxytoca, and E. faecalis) and methods above. One milliliter of PC was injected intraperitoneally using a 1 mL insulin syringe (BD). The optimal and lethal dose of PC was determined by preliminary experiments. FMT and AC-FMT were prepared as described above. Each mouse received two doses of either FMT, AC-FMT immediately after i.p. injection and 14 h later. Survival and health status were monitored and documented for 72 h using the clinical scoring system shown in Supplementary Table 2 (mice that survive until this point, regardless of treatment group return to full health and survive indefinitely). All mice that were moribund were sacrificed and considered to be a mortality.

**Quantitative polymerase chain reaction (qPCR).** RNA was isolated from cecum, liver, and spleen as before and converted to cDNA using reverse transcriptase (Promega) according to the manufacturer's instructions. Expression analysis was performed in duplicate via qPCR on a Roche LightCycler 480 using SYBR Advantage qPCR Premix (Clontech).

Expression levels were quantified and normalized to housekeeping genes HPRT (for qPCR on organs in both mouse models), GAPDH (for in vitro MEF experiments), or ASL (for 16S relative expression). For Fig. 3b, Supplementary Fig. 3a, and Fig. 4d, relative expression values were normalized to the mean of the untreated controls for each organ and displayed as fold change to allow for display on the same axis. For Fig. 3c, Supplementary Fig. 3b, c, and Fig. 4b, relative expression values were normalized to untreated MEF baseline relative expression for each gene (indicated with a dotted line at 1) and displayed as fold change. For Supplementary Fig. 4a, the number of 16S copies was calculated using a purified Ruminococcus productus DNA standard and normalized to the relative expression of host ASL for each mouse assessed[61]. The qPCR primers shown in Supplementary Table 4 were custom ordered from Integrated DNA Technologies.

**MEF and PC co-culture.** C57BL/6 primary MEFs were obtained from 12.5 to 14.5 days post coitus embryos and cultured in Dulbecco's Modified Eagle's Medium (Gibco) supplemented with 10% fetal bovine serum (Gibco), and 1% non-essential amino acids (Gibco). MEFs were plated at a density of $2.5 \times 10^5$/mL in 24-well plates (Costar) overnight. Live PC members were prepared as before to an OD600 of 0.2, and diluted (1/$10^5$ by volume) in MEF containing culture media. PC lysates were prepared by taking live PC culture, bead beating with 0.1-mm diameter glass beads (BioSpec) for 5 min and filtering using Millex-GV 0.22-μm filter (Millipore). Plates were incubated at 37 °C, 5% $CO_2$ for 1, 3, 6, and 14 h for live PC, or 12 h for lysates in the presence of acetic acid (Fisher), butyric acid (Sigma), propionic acid (Sigma), nicotinic acid (Sigma), or valproic acid sodium salt (Sigma) at the indicated concentrations. After incubation, culture media was removed and cells were harvested in Buffer RLT Plus (Qiagen) with 1% 2-Mercaptoethanol (Sigma) and RNA extraction was performed using the RNeasy Plus Mini Kit (Qiagen) according to the manufacturer's instructions. All MEF experiments were ≥2 independent experiments, with at least two experimental replicates/experiment. The mean fold change for each experiment was plotted and represented as bar graphs.

**Gas chromatography–mass spectrometry.** SCFA were extracted from mouse cecal contents using diethyl ether (Fisher Scientific), derivatized using N-tert-Butyldimethylsilyl-N-methyltrifluoroacetamide with 1% tert-Butyldimethylchlorosilane (Sigma) and run on an Agilent Single Quad GC–MS (5977A Single Quad and 7890B GC). All values are normalized to fecal sample mass. Raw spectral data were uploaded to the MassIVE repository (https://doi.org/10.25345/C5JQ3V).

**Statistical analysis.** Statistical analysis was performed using GraphPad Prism software. Unpaired t-test (two-tailed) was used when analyzing the differences between two means, whereas one-way ANOVA/Tukey's multiple comparisons test was used when more than two means were compared. To determine the statistical significance of quantitative culture analyses, we used nonparametric tests because of the distribution of data: Mann–Whitney test (two-tailed) was used to when two groups were compared, Kruskal–Wallis test/Dunn's multiple correction test was used for three groups. Log-rank (Mantel–Cox) test was used to determine statistical significance between Kaplan–Meier survival curves.

**Reporting summary.** Further information on research design is available in the Nature Research Reporting Summary linked to this article.

## Data availability

The datasets generated during and/or analyzed during the current study are available in the Article, Supplementary Information files, or available from the corresponding author on reasonable request. Microarray data was uploaded to the NCBI Gene Expression Omnibus database, GEO accession number: GSE71530. Genome sequences of the ICU1-2 pathogen community members were uploaded to NCBI in BioProject PRJNA307050. Raw spectral data for measurement of SCFA by GC–MS were uploaded to the MassIVE repository [https://doi.org/10.25345/C5JQ3V].

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

## Acknowledgements

Microarray sequencing was performed by the University of Chicago Genomics Facility. 16S sequencing was performed by the Environmental Sample Preparation and Sequencing Facility at Argonne National Laboratory. The data reported in this paper are tabulated in the Supplementary Materials and archived in NCBI. We thank Dr Tatyana V. Golovkina (UChicago) for IRF3$^{-/-}$ mice. We thank Dr Toufic Mayassi (UChicago), and Dr Brad. A Palanski (Stanford) for helpful discussions and Dr Reinhard Hinterleitner (UChicago), and Dr Christopher M. Carmean (University of Illinois Chicago) for critical reading of the paper. This study was funded by NIH RO1 5R01GMO62344 (JCA), NIH 5P30DK042086-27 (BJ), NIH T32GM007281 (KS), and the Virginia and D. K. Ludwig Fund for Cancer Research (NNK, DRR, and RRW).

## Author contributions

JCA conceived of the overall study, and BJ and JCA designed the studies. OZ, BJ, JCA directed the study and guided the interpretation of the results. SMK, and JRD designed experiments and analyzed the results. SMK, JRD, SKH, KS, RK, MAK, KGH, AM, AZ, RB, DRR, MW, JDE, BAS, and IDF performed the experiments. NNK and LBB

performed transcriptional data analysis. NS, SC, DAA, and JAG performed 16S analysis. DRR and RRW provided essential reagents. SMK, BJ, and JCA wrote the paper.

## Competing interests

The authors declare no competing interests.
