## [Peer Review File · Nature Communications]

Reviewers' comments:

Reviewer #2 (Remarks to the Author):

The authors have answered the criticisms and the questions I had and the manuscript is substantially improved. It is a very important paper. Alexander Khoruts, MD

Reviewer #3 (Remarks to the Author):

The manuscript represents a transfer from a sister journal where I previously reviewed the first submission. The authors have responded to my comments, however there are only very few additional experiments to answer any of the questions, although I find it hard to compare the documents without a clearly marked revised version. I agree with the authors that the level of mechanistic insight can be lower at this stage, however I still stand with the following comments:

1. The selection of just a few "microarray mice" is problematic, the authors state in their response that they -by 25 years of clinical experience- selected the most contrasting mice for analysis. Although I disagree with that procedure, it should at least be clearly noted in the M&M section.
2. The microbiota analysis as it stands in the main part of the manuscript is below standard, respective suppl data figures are informative and necessary to go into the main part, with depiction of statistical differences (not just in another supplemental table). The respective inocula must be included in the analysis and not just in the rebuttal letter.
3. The IRF3 story is has remained more or less unaltered, as differential roles for IRF3 in different cell types have been reported, I still find the use of MEFs as a model system a very strange selection. The authors do not discuss the fact that IRF3 KO has been reported to protective in several sepsis models including CLP. I agree that different model systems can lead to different outcomes however this must be openly discussed.
4. The pIRF3/IRF3 protein level evidence is underwhelming, as it shows n=1 and just ubiquitous brown staining in liver /SI. In the current form this cannot support the claim and must be removed.

5. I still stand with the notion that the term MDR healthcare pathogen infection is confusing, as it suggests that MDR pathogens relevant to healthcare can be reduced to 3 bacteria and Candida isolated from a single patient. In fact, the Candida strain shows no resistance to antifungals and the Enterococcus strain is sensitive to most clinically applied antibiotics including Vanco. I would give a strong advice to thus change the nomenclature of the model.

6. The additional OTU analysis is very interesting, however the rationale and exact algorithm is not sufficiently described in the manuscript , is this based on an indicator OTU analysis , how did the authors set the thresholds for OTU identification ? The Figure itself contains very small and repetitive text, with strange technical lingo and abbreviations , this should be simplified.

Reviewer #3 (Remarks to the Author):

The manuscript represents a transfer from a sister journal where I previously reviewed the first submission. The authors have responded to my comments, , however there are only very few additional experiments to answer any of the questions, although I find it hard to compare the documents without a clearly marked revised version. I agree with the authors that the level of mechanistic insight can be lower at this stage, however I still stand with the following comments:

We thank the reviewer for all of their suggestions. We have highlighted all the changes that have made in this iteration of the manuscript for ease of comparison.

1. The selection of just a few "microarray mice" is problematic, the authors state in their response that they -by 25 years of clinical experience- selected the most contrasting mice for analysis. Although I disagree with that procedure, it should at least be clearly noted in the M&M section.

As requested by the reviewer, the Materials and Methods section has been expanded to include the following explanation of how mice were chosen for microarray analysis:

Mouse genome transcription analysis

As indicated in the original model description, mice underwent the full protocol to allow for the development of gut derived sepsis (i.e. intestinal inoculation of the pathogen community and surgical hepatectomy) and then were treated with either a live FMT or AC-FMT. Within 24-48 hours, mice declare themselves in a very clinically apparent way whether they are healthy and clearly going to recover (moving frequently, jumped to touch, eating, grooming, stooling, etc.) or they appear grossly clinically septic (lethargy, ruffled fur, chromodacryorrhea, lying on side, do not move to touch). Using these criteria, mice were chosen from the corresponding FMT or AC-FMT treated groups (given the clear differentiation of each in terms of both sepsis development and mortality) and their tissues submitted for microarray analysis.

2. The microbiota analysis as it stands in the main part of the manuscript is below standard, respective suppl data figures are informative and necessary to go into the main part, with depiction of statistical differences (not just in another supplemental table). The respective inocula must be included in the analysis and not just in the rebuttal letter.

For the gut-derived sepsis model, we chose to place the microbiota analysis in Supplementary Materials because we observed restoration of microbial diversity and clearance of pathogens in a manner that is consistent with previous reports (references 10, 11, 12). For the IP model, we include all microbiome readouts that could account for the differences that we see in survival and pathogen clearance in the main manuscript (Figure 4a,b). As suggested, we have integrated statistical comparisons into

Supplementary Figure 2a, and added the 16S rRNA gene sequencing of the inoculum into Supplementary Figure 5c.

3. The IRF3 story is has remained more or less unaltered, as differential roles for IRF3 in different cell types have been reported, I still find the use of MEFs as a model system a very strange selection. The authors do not discuss the fact that IRF3 KO has been reported to protective in several sepsis models including CLP. I agree that different model systems can lead to different outcomes however this must be openly discussed.

We thank the reviewer for this comment, and have added the following statement and reference into our revised version of the manuscript:

These results are in contrast to findings reported with the cecal puncture and ligation model of sepsis in which commensal microbiota and a grossly ischemic cecum drive the process of lethal sepsis. In the cecal ligation model, IRF3-deficiency has been shown to be protective³⁰, potentially indicating the difference between sepsis driven by commensal microbiota versus healthcare-associated pathogens.

30. Walker, W. E., Bozzi, A. T. & Goldstein, D. R. IRF3 contributes to sepsis pathogenesis in the mouse cecal ligation and puncture model. *J. Leukoc. Biol.* **92**, 1261–1268 (2012).

4. The pIRF3/IRF3 protein level evidence is underwhelming, as it shows n=1 and just ubiquitous brown staining in liver /SI. In the current form this cannot support the claim and must be removed.

We have removed the pIRF/IRF3 staining from the final version of our manuscript.

5. I still stand with the notion that the term MDR healthcare pathogen infection is confusing, as it suggests that MDR pathogens relevant to healthcare can be reduced to 3 bacteria and *Candida* isolated from a single patient. In fact, the *Candida* strain shows no resistance to antifungals and the *Enterococcus* strain is sensitive to most clinically applied antibiotics including Vanco. I would give a strong advice to thus change the nomenclature of the model.

We have changed the nomenclature for our pathogen community from “MDR healthcare pathogens” to “human healthcare-associated pathogens” in the title and throughout the manuscript to reflect the fact that not all four pathogen community members have multi-drug resistance.

6. The additional OTU analysis is very interesting, however the rationale and exact algorithm is not sufficiently described in the manuscript , is this based on an indicator

OTU analysis , how did the authors set the thresholds for OTU identification ? The Figure itself contains very small and repetitive text, with strange technical lingo and abbreviations , this should be simplified.

We appreciate the reviewer's comments on the additional OTU analysis presented in Figure 4b and Supplementary Figure 6. We have included additional text in the figure legend to describe the rationale and programs used to analyze the data. In brief, the R packages phyloseq and DESeq2 were used to identify the differential abundances of OTUs as pairwise comparisons of the datasets. Similarity scores (Sab) for OTU identification based on the RDP SeqMatch tool are listed parenthetically on the right-hand side of the figure. The highest scoring genus-species designation is listed. We also simplified the content of the Figure 4b and Supplementary Figure 6 by collapsing repetitive listings of the taxonomic classifiers on the left-hand side, and genus names on the right-hand side.